# MiRNAs in Lung Adenocarcinoma: Role, Diagnosis, Prognosis, and Therapy

**DOI:** 10.3390/ijms241713302

**Published:** 2023-08-27

**Authors:** Yongan Song, Leonardo Kelava, István Kiss

**Affiliations:** 1Department of Public Health Medicine, University of Pécs Medical School, Szigeti Str. 12, 7624 Pécs, Hungary; 2Department of Thermophysiology, Institute for Translational Medicine, Medical School, University of Pécs, Szigeti Str. 12, 7624 Pécs, Hungary

**Keywords:** miRNA, LUAD, tumor suppressor, oncogenic, diagnosis, prognosis, therapy

## Abstract

Lung cancer has emerged as a significant public health challenge and remains the leading cause of cancer-related mortality worldwide. Among various types of lung malignancies, lung adenocarcinoma (LUAD) stands as the most prevalent form. MicroRNAs (miRNAs) play a crucial role in gene regulation, and their involvement in cancer has been extensively explored. While several reviews have been published on miRNAs and lung cancer, there remains a gap in the review regarding miRNAs specifically in LUAD. In this review, we not only highlight the potential diagnostic, prognostic, and therapeutic implications of miRNAs in LUAD, but also present an inclusive overview of the extensive research conducted on miRNAs in this particular context.

## 1. Introduction

For several decades, lung cancer has sustained its position as the primary cause of cancer-related mortality, presenting a significant global public health challenge. It is histopathologically categorized into small cell lung cancer (SCLC) and non-small cell lung cancer (NSCLC), with NSCLC accounting for over 80% of cases. NSCLC includes adenocarcinoma (LUAD), squamous cell carcinoma (LUSC), and large cell carcinoma (LCC), with LUAD being the most prevalent subtype, comprising around 40–45% of cases [1]. Despite recent advancements in detection methods and targeted therapies, the majority of LUAD cases are diagnosed at advanced stages and have a poor prognosis, with a 5-year survival rate of only around 15% [2]. This grim prognosis is due to the late-stage disease presentation, heterogeneous tumor characteristics with a variety of histological subtypes, and our insufficient understanding of tumor biology. The development of targeted therapies that address specific molecular alterations necessitates precise subclassification of lung cancer, a task that surpasses the capabilities of standard histopathological diagnostic techniques. Moreover, due to our incomplete comprehension of the pathogenesis and dynamic fluctuations in tumor gene expression profiles, the progress of LUAD therapy has reached a plateau, demanding urgent breakthroughs [3]. Therefore, genomic medicine emerges as a promising field in complementing and expanding LUAD research.

MicroRNAs (miRNAs) are small non-coding RNAs that were first identified in 1993 during studies of *Caenorhabditis elegans*. It was quickly recognized that these seemingly conserved miRNA sequences play a crucial role in regulatory pathways in eukaryotes [4]. MiRNAs can interfere with mRNA translation through complementary base pairing with the 3′ untranslated region (UTR) of target mRNAs, leading to either mRNA degradation or translational repression [5]. Around 70% of miRNAs are transcribed from specialized miRNA loci, while the remaining fraction of miRNAs is processed from introns of protein-coding genes. In most cases, miRNA genes are transcribed in the nucleus by RNA polymerase II (Pol II). This leads to the formation of primary miRNA (pri-miRNA) that undergoes capping, splicing, and polyadenylation [6]. One pri-miRNA can generate either a single miRNA or a cluster containing two or more miRNAs. These long pri-miRNAs require cleavage by a microprocessor complex, primarily composed of the double-stranded RNase III enzyme DROSHA and the double-stranded RNA (dsRNA) binding protein DiGeorge syndrome critical region 8 (DGCR8) [7]. The microprocessor cleaves one strand of the dsRNA at the base of the stem-loop secondary structure within the pri-miRNA, releasing a hairpin-shaped precursor miRNA (pre-miRNA) of approximately 60–70 nucleotides [8,9].

While the core components of the microprocessor, DROSHA and DGCR8, are essential for the biogenesis of almost all miRNAs in cells, there are several cofactors that also play a role in this process. The pre-miRNA is exported from the nucleus to the cytoplasm by the exporter protein 5 (XPO5) and subsequently processed by DICER1, an RNase III enzyme that cleaves the pre-miRNA at both its 5′ and 3′ ends [10,11]. The pre-miRNA is cleaved into miRNA duplexes, and one of the strands is selected as the mature miRNA, which is loaded into the RNA-induced silencing complex (RISC) to act as a negative regulator of gene expression. The other strand is eventually degraded (Figure 1). The degradation or translational inhibition of target mRNAs by miRNAs largely depends on the complementarity between the miRNA’s 5′-seed sequence and the mRNA’s 3′-UTR element. Moreover, defects in the miRNA biogenesis machinery may contribute to tumorigenesis [12]. Multiple lines of evidence have demonstrated that miRNAs have diverse cellular regulatory roles, with some miRNAs being recognized as oncogenes or tumor suppressor genes [13]. Numerous studies have shown that human cancers, including LUAD, exhibit a large number of dysregulated miRNAs. Therefore, these miRNAs may serve as potential diagnostic or prognostic markers and could even guide therapeutic interventions [14]. MiRNAs can be utilized to subclassify tumors, as miRNA expression profiles serve as powerful indicators of pathological parameters and reliable biomarkers in LUAD [15].

Numerous public databases of miRNAs have been established, which accumulate data on various aspects of thousands of annotated human miRNAs, including an increasing number of miRNAs associated with LUAD [16]. There is a growing interest in identifying and characterizing miRNAs from different types of body fluids due to the ease of access to these samples. MiRNAs can also be obtained from tissue samples. Particularly, miRNAs derived from formalin-fixed and paraffin-embedded (FFPE) samples exhibit greater resistance to degradation compared to mRNAs [17]. Consequently, these samples stored in hospitals have a significant advantage for miRNA research.

On the other side, while miRNAs are an exciting new target for treatment of cancer, adaptation of miRNAs for therapy presents a challenge because of the lack of specificity. One miRNA typically targets a cluster of genes, so manipulating its expression can bring about undesired consequences.

In this review, we not only highlight the potential diagnostic, prognostic, and therapeutic implications of miRNAs in LUAD, but also present an inclusive overview of the extensive research conducted on miRNAs in this particular context. We believe that our review will facilitate researchers in comprehending the advancements made in the study of miRNAs in LUAD. Additionally, we have compiled an appendix encapsulating nearly all miRNAs pertinent to the development and treatment of LUAD, facilitating an easy reference and further investigation into the correlation between specific miRNAs and LUAD.

## 2. Tumor Suppressor and Oncogenic miRNAs in LUAD

### 2.1. Tumor Suppressor miRNAs in LUAD

In LUAD research, *let-7*, *miR-34*, and *miR-200* are generally acknowledged as classical and substantial miRNAs. The *let-7* family can regulate the timing of stem cell division, differentiation, and apoptosis. Dysregulation of *let-7* can result in a less differentiated cellular state, which is closely associated with the development of cell-based diseases, including cancer. Notably, the *let-7* family was the first miRNA identified with reduced expression in lung cancer [18]. When compared to normal lung tissue, *let-7* expression decreased in 79% of adenocarcinomas (adenocarcinomas in situ and well-differentiated invasive adenocarcinomas), suggesting its role in early-stage carcinogenesis [19]. Further investigations have revealed the therapeutic potential of *let-7a* overexpression. In A549 cells, *let-7a* overexpression effectively suppresses cancer cell proliferation, migration, and invasion. Additionally, *let-7a* induces apoptosis and cell cycle arrest by modulating the cyclin D1 signals [20]. Both in vitro and in vivo studies have shown that *let-7b-3p* inhibits LUAD cell proliferation and metastasis by targeting the TFIIB-related factor 2 (BRF2)-mediated MAPK/ERK pathway [21]. Likewise, *let-7c* has shown anti-proliferative effects in LUAD. By inhibiting TRIB2, *let-7c* increases the activity of downstream signals, namely, C/EBP and p38MAPK, effectively suppressing the proliferation of A549 cells in both in vitro and in vivo models [22]. Moreover, the *let-7* family targets the *RAS* gene family, notably *KRAS*, a critical oncogene in LUAD. However, it is noteworthy that polymorphism in the *KRAS* 3‘UTR, affecting its binding to *let-7*, does not appear to impact patient 5-year survival [23].

The *miR-34* family is also a canonical miRNA family in LUAD, comprising three members (*miR-34a*, *miR-34b*, and *miR-34c*) clustered at two distinct chromosomal loci, *Mir34a and Mir34b/c*. In a syngeneic mutant mouse model, both *miR-34a* and *miR-34b/c* can block lung metastasis, with *miR-34b/c* exhibiting a stronger tumor growth suppression effect than *miR-34a*. *MiR-34b/c* also reduces the expression of mesenchymal markers (cadherin-2 and fibronectin) and increases the expression of epithelial markers (claudin-3 and desmoplakin) compared to *miR-34a*. Deletion of all three *miR-34* family members promotes mutated *KRAS*-driven lung tumor progression in mice, and in LUAD patients, higher expression of *miR-34a/b/c* is associated with better survival [24]. There is no difference in the expression of *miR-34a* between metastatic and non-metastatic LUAD. However, for *miR-34b* and *miR-34c*, the expression levels are significantly lower in metastatic LUAD than in non-metastatic LUAD, suggesting a correlation of low expression with distant metastasis in LUAD. Additionally, promoter hypermethylation of *miR-34a* and *miR-34b/c* is a common event in LUAD [25]. Sato et al. transfected *miR-34b* into A549 and PC9 cell lines, resulting in suppressed lung cancer cell proliferation and IGBP1 expression. Furthermore, *miR-34b* transfection induced apoptosis in LUAD cell lines, similar to the effect of siIGBP1 RNA [26]. A positive feedback loop exists between *p53* and *miR-34* to mediate tumor suppression. Okada et al. demonstrated that *miR-34a* inhibits HDM4, a potent negative regulator of *p53*, creating a positive feedback loop on *p53*. In a *KRAS*-induced mouse lung cancer model, the absence of miR-34a alone does not exhibit strong oncogenic effects. However, *miR-34a* deficiency strongly promotes tumorigenesis when combined with *p53* haploinsufficiency, suggesting that a defective *p53-miR-34* feedback loop can enhance tumorigenesis under specific circumstances [27].

*MiR-200* is a highly influential miRNA in LUAD, playing a major role in the epithelial-mesenchymal transition (EMT). In a genetic mouse model of metastatic LUAD, forced expression of *miR-200* in metastasis competent-cells injected in syngeneic mice abolished their ability to undergo EMT, invasion, and metastasis, and conferred transcriptional features of metastasis-incompetent tumor cells [28]. Schliekelman et al. have identified ECM proteins and peptidases directly regulated by *miR-200* and revealed that *miR-200* expression can alter the tumor microenvironment to suppress EMT and metastatic processes [29]. The *miR-200* family (*miR-200a/b/c*) mediates a key EMT step in LUAD development by regulating the expression of ZEB1 and E-cadherin (CDH1) [30]. Additionally, *miR-200* downregulates BMP4 by directly targeting the GATA4 and GATA6 transcription factors. BMP4 upregulates JAG2, an upstream factor of *miR-200*, and the JAG2-*miR-200*-BMP4 loop is involved in regulating LUAD cell growth, migration, invasion, tumorigenesis, and metastasis in syngeneic mice [31]. BMP4 also enhances the expression of the T cell co-inhibitory receptor ligand PD-L1 in mesenchymal subsets of LUAD cells, resulting in CD8^+^ T cell-mediated immunosuppression that promotes tumor growth and metastasis [32]. Moreover, deficiency of *miR-200* in LUAD cells promotes the proliferation and activation of adjacent cancer-associated fibroblasts (CAFs), thereby enhancing the metastatic potential of cancer cells. *MiR-200* regulates the functional interaction between cancer cells and CAFs by targeting the Notch ligands Jagged1 and Jagged2 in cancer cells and inducing Notch activation in adjacent CAFs. Thus, interactions between cancer cells and CAFs constitute an important mechanism for promoting metastatic potential [33]. Notch ligand Jagged2 promotes LUAD metastasis in mice through a *miR-200*-dependent pathway [34]. Overexpression of *miR-200a* also blocks the increase in LUAD cell proliferation caused by GOLM1 overexpression [35]. Additionally, *miR-200* can activate AKT in LUAD cells to promote cell proliferation through a FOG2-independent mechanism involving IRS-1 [36].

In addition to the classic tumor suppressor miRNAs, numerous miRNAs have been found to be closely related to LUAD. Hundreds of miRNAs have been shown to inhibit LUAD proliferation, enhance drug sensitivity, and suppress migration. For instance, transfection of *miR-145* into epidermal growth factor receptor (EGFR) mutant LUAD cells led to a significant inhibition of cell growth [37]. *MiR-145* targets EGFR and nucleoside diphosphate-linked partial X-type motif 1 (NUDT1 or MTH1) in LUAD cells, thereby inhibiting cell proliferation [38]. *MiR-206* and *miR-140* act as tumor suppressors, modulating oncogenic TRIB2 promoter activity through p-Smad3, inducing LUAD cell death, and inhibiting cell proliferation [39]. In cultured LUAD cells treated with recombinant TGF-β, ectopic expression of *miR-206* impaired canonical signaling and the expression of TGF-β target genes associated with epithelial-mesenchymal transition, partially due to the suppression of Smad3 protein levels in LUAD cells expressing ectopic *miR-206* [40]. *MiR-29a* can inhibit the growth, migration, and invasion of LUAD cells by targeting CEACAM6. *MiR-29c* also plays a tumor suppressor role in LUAD by directly binding to the 3′-UTR of vascular endothelial growth factor A (VEGFA) and repressing its expression. Furthermore, VEGFA regulated by *miR-29c* is biologically active and affects HUVEC (human umbilical vein endothelial cell) tube formation. Transfection with VEGFA expression plasmid counteracted the effects of the *miR-29c* mimics. The association of *miR-29c* with microvessel density (MVD) and VEGFA was further confirmed in patient samples [41]. Overexpression of *miR-576-3p* inhibited LUAD cell migration and invasion, decreased the expression of mesenchymal markers, and directly targeted serum/glucocorticoid-regulated kinase 1 (SGK1). Modulation of *miR-576-3p* levels resulted in changes in SGK1 protein and mRNA and activation of downstream targets associated with metastasis [42]. *MiR-520c-3p*, by targeting AKT1 and AKT2, regulates multiple biological functions and cellular behaviors, and inhibits the proliferation, migration, and invasion of LUAD [43]. Downregulation of the *miR-150* duplex has been observed in clinical specimens of LUAD, and *miR-150-3p* has been found to directly regulate TNS4 expression in LUAD cells. Aberrant expression of TNS4 has been detected in clinical specimens of LUAD, and its aberrant expression has been shown to increase the invasiveness of LUAD cells [44]. All members of the *miR-143/miR-145* cluster act as tumor suppressor miRNAs in LUAD. They may control many genes responsible for LUAD malignancy [45]. Moreover, AKT2 affects the proliferation, migration, and invasion of LUAD cells by regulating the cell cycle, promoting the occurrence of EMT, and the expression of matrix metalloproteinases (MMPs). Overexpression of *miR-124* can downregulate AKT2 to inhibit LUAD development and progression in vivo and in vitro [46]. Some important tumor suppressor miRNAs are listed in Table 1. We have compiled almost all tumor suppressor miRNAs in Appendix A.

### 2.2. Oncogenic miRNAs in LUAD

*MiR-21*, a well-known oncogenic miRNA, is upregulated in various types of cancer. In LUAD tissues, the expression of *miR-21-5p* was found to be significantly higher than in normal tissues. Additionally, high expression of *miR-21-5p* was associated with a poorer prognosis, with a hazard ratio (HR) of 1.59 compared to the low expression group [47]. Several studies have investigated the molecular mechanisms by which *miR-21* promotes LUAD. One of these mechanisms involves the downregulation of WWC2 expression by *miR-21-5p*. Silencing *miR-21-5p* or overexpressing WWC2 inhibited PC9 cell proliferation, migration, and invasion. Western blot analysis showed that overexpression of WWC2 hindered the EMT process in LUAD cells [48]. Furthermore, *miR-21* inhibits the Hippo signaling pathway by targeting KIBRA, thus promoting the progression of LUAD [49]. Additionally, SET is a direct target of *miR-21-5p*. Stable knockdown of *miR-21-5p* significantly enhanced SET/TAF-Iα expression and inhibited A549 cell migration, invasion, proliferation, and tumorigenicity [50]. In addition to lung cancer cells, *miR-21* expression in lung fibroblasts may trigger their transdifferentiation into cancer-associated fibroblasts, inducing a novel CAF-secreted protein called calumenin, as well as known CAF markers such as periosteal protein, α-smooth muscle actin, and podoplanin, thereby supporting cancer progression [51]. Moreover, changes in *miR-21* expression were more pronounced in cases with EGFR mutations. A strong correlation was observed between phosphorylated-EGFR (p-EGFR) and *miR-21* levels in LUAD cell lines, and the inhibition of *miR-21* by EGFR-TKI and AG1478 suggested that EGFR signaling is a pathway that positively regulates *miR-21* expression. Antisense inhibition of *miR-21* enhanced AG1478-induced apoptosis in the LUAD cell line H3255 with mutant EGFR and high levels of *miR-21*. Antisense *miR-21* can induce apoptosis alone and also enhance the effect of AG1478 in H441 cells with EGFR wild-type. Abnormally increased *miR-21* expression, further enhanced by activated EGFR signaling, plays an important role in LUAD development in both never-smokers and smokers [52].

*MiR-31* is another significant oncogenic miRNA. *MiR-31-5p* has been found to be significantly upregulated in LUAD tissues and cell lines. Its overexpression has been shown to promote cell proliferation and migration while inhibiting apoptosis. *MiR-31-5p* can directly target TNS1 to promote LUAD cell growth through the TNS1/*p53* axis [53]. In a transgenic mouse model of LUAD, induced expression of *miR-31* synergizes with mutated *KRAS* to accelerate lung tumorigenesis. *MiR-31* has been identified as a regulator of lung epithelial cell growth, targeting six negative regulators of RAS/MAPK signaling. Mick D Edmonds has distinguished *miR-31* as a driver of lung tumorigenesis, promoting mutant *KRAS*-mediated tumorigenesis by directly reducing the expression of these negative regulators of RAS/MAPK signaling [54]. Additionally, Zeb1 has been found to regulate the symmetrical division of mouse Lewis LUAD stem cells through miR-31-mediated Numb [55].

Recent studies have identified an increasing number of oncogenic miRNAs in LUAD, revealing their regulatory roles in this disease. One such miRNA, *miR-708*, was found to be highly expressed in most LUAD cases, including those from both smokers and non-smokers. Overexpression of *miR-708* was found to lead to increased cell proliferation, migration, and invasion, making it an oncogene that promotes tumor growth and disease progression. This effect is achieved by directly downregulating TMEM88, a negative regulator of the Wnt signaling pathway. Patients with LUAD who had low expression of *miR-708* tended to have better survival than those with high expression [56]. Another oncogenic miRNA, *miR-150*, was shown to inhibit SRCIN1, leading to the activation of the Src/focal adhesion kinase (FAK) and Src/Ras/extracellular signal-regulated kinase (ERK) pathways, ultimately promoting the proliferation and migration of A549 cells [57]. *MiR-483-5p* promotes EMT associated with invasiveness and metastasis in LUAD. It is activated by the WNT/β-catenin signaling pathway and exerts its metastasis-promoting effect by directly targeting two metastasis suppressors, Rho GDP dissociation inhibitor α (RhoGDI1), and activated leukocyte adhesion molecule (ALCAM). Downregulation of RhoGDI1 enhances the expression of Snail, thereby promoting EMT [58]. *MiR-297* was found to be upregulated in LUAD compared with adjacent normal tissues, as well as in tested LUAD cell lines. Ectopic expression of *miR-297* enhances LUAD cell proliferation and colony formation and promotes cell migration and invasion. Glypican-5 (GPC5) was identified as a direct target gene of *miR-297* in LUAD cells [59]. Pro-inflammatory signals, such as tumor necrosis factor (TNF), promote metastasis in LUAD by regulating *miR-146a* and reducing the expression of the tumor suppressor protein Merlin. Furthermore, invasive and metastatic tumors in humans had higher levels of TNF and *miR-146a* but lower levels of Merlin protein compared to non-invasive tumors. TNF-induced upregulation of *miR-146a* in LUAD promotes repression of Merlin protein and subsequent metastasis [60]. We have listed some important oncogenic miRNAs in Table 2 and included almost all oncogenic miRNAs in Appendix A.

## 3. MiRNA Sponges: Competitive Endogenous RNAs (ceRNAs)

MiRNAs have been demonstrated to regulate gene expression by binding to target mRNAs, leading to translational repression or mRNA degradation. Moreover, networks consisting of protein-coding and noncoding RNAs, including long noncoding RNAs (lncRNAs), pseudogenes, small noncoding RNAs, and circular RNAs (circRNAs), can compete for the limited pool of miRNAs. These RNA molecules are referred to as competing endogenous RNAs (ceRNAs) and have the ability to sequester miRNAs, serving as natural miRNA sponges. Consequently, these RNAs co-regulate each other within a complex ceRNA network to suppress miRNA activity [61,62]. Extensive research has been conducted on miRNAs in LUAD, and the role of ceRNA sponges has become a subject of further in-depth investigation.

LncRNAs are noncoding RNAs longer than 200 base pairs [63]. According to the ceRNA hypothesis, lncRNAs can act as sponges for miRNAs, weakening their influence on mRNA. In LUAD, *LINC00466* promotes tumorigenesis, invasion, migration, and proliferation, while inhibiting apoptosis by suppressing the expression of *miR-144* [64]. Another lncRNA, *FBXL19-AS1*, functions as a *miR-203a-3p* sponge. The *FBXL19-AS1/miR-203a-3p* axis regulates LUAD cell metastasis by targeting downstream genes, including survivin, E2F1, and ZEB2 [65]. *VPS9D1-AS1* promotes malignant progression in LUAD cells by acting as a *miRNA-30a-5p* sponge, and KIF11 is a downstream target of *miRNA-30a-5p*. *VPS9D1-AS1* upregulates KIF11 expression by competitively sponging *miRNA-30a-5p*, and KIF11 restores the effect of *miRNA-30a-5p* on LUAD cells [66]. Additionally, the lncRNA *SNHG7* is downregulated in LUAD tissues compared to normal tissues. *SNHG7* interacts with *miRNA-181* and upregulates the tumor suppressor cbx7, which inhibits the Wnt/β-catenin pathway in LUAD [67]. Knockdown of *LINC00960* inhibits the proliferation, migration, and invasion of LUAD cells by acting as a *miR-124a* sponge to inhibit the SphK1/S1P pathway. The interaction between *miR-124a* and *LINC00960* or SphK1 was confirmed by luciferase reporter and RNA pull-down assays [68].

CircRNAs also play important roles in the regulation of cancer and are involved in ceRNA networks. *Circ-CAMK2A* targets *miR-615-5p*, leading to increased expression of fibronectin 1 by sponging *miR-615-5p*. This increase, in turn, promotes the expression of MMP2 and MMP9, facilitating the metastasis of LUAD [69]. High expression of *hsa_circ_0000326* correlates with tumor size, regional lymph node status, and differentiation in human LUAD. *Hsa_circ_0000326* enhances cell proliferation and migration while inhibiting apoptosis. It functions as a competitive binding agent for *miR-338-3p*, modulating its activity and subsequently upregulating the expression of the downstream target RAB14 [70]. *Has_circ_0001588* upregulates the expression of NACC1 by binding to *miR-524-3p* and promotes the proliferation, migration, and invasion of LUAD cells [71]. The oncogenic role of *circCSNK1G3* was inferred from its aberrant expression and its association with enhanced proliferation, migration, and invasion in A549 and H1299 cells. It induces HOXA10 expression, promoting the growth and metastasis of LUAD cells through the sponging of *miR-143-3p* [72]. Studies have shown that *circ_0001361* promotes the growth and metastasis of LUAD cells by acting as a sponge for *miR-525-5p*, which upregulates the downstream target VMA21 levels. Inhibition of *circ_0001361* suppresses xenograft tumor growth in vivo by modulating the *miR-525-5p*/VMA21 axis [73]. Clinical samples obtained from LUAD surgery have shown that the expression of *circMMD_007* is abnormally elevated, particularly in late stages of LUAD. Knockdown of *circMMD_007* blocks LUAD initiation in vitro and in vivo by negatively regulating the expression of *miR-197-3p*. PTPN9 appears to be a molecular target of *miR-197-3p* [74].

These studies highlight the promising role of miRNAs in LUAD through the ceRNA network, making them potential candidates for future RNA-based research advancements in this disease. Table 3 presents some ceRNA networks, and Appendix A contains a comprehensive list of miRNAs involved in ceRNA networks in LUAD.

## 4. MiRNAs as Diagnostic Biomarkers in LUAD

Due to the complexity of lung cancer subtypes and limited understanding of risk factors, lung cancer is often diagnosed at an advanced stage, resulting in limited treatment options. Recent advancements in cancer treatment emphasize the need for accurate histological subtyping during diagnosis to optimize therapeutic response and minimize adverse effects. Early diagnosis and personalized treatment options are crucial for improving clinical outcomes in LUAD [75]. However, challenges such as interobserver variability, tumor heterogeneity, and variations in degree of differentiation can impact the pathological diagnosis of lung cancer. Relying solely on morphological assessment may be inadequate for precise distinction. Therefore, the utilization of gene signatures may play a vital role in facilitating faster diagnosis and classification. Numerous studies have focused on identifying and characterizing miRNA expression signatures in LUAD. In addition to the miRNAs mentioned in the following section, Table 4 includes additional promising miRNAs for the diagnosis of LUAD.

### 4.1. Tissue Sample

Lung cancer tissue and adjacent normal tissue are commonly used samples for research. The *let-7* family of miRNAs has been identified as highly differentially expressed between LUAD and LUSC. Notably, these differences in histological expression are most prominent in early-stage tumors rather than advanced tumors. Therefore, leveraging *let-7* differences for mechanistic insights or therapeutic benefits should primarily focus on early-stage tumors [76]. In a study using the same test samples along with 88 additional validation samples, the utility of three specific miRNAs (*miR-196b*, *miR-205*, *and miR-375*) as biomarkers to distinguish between LUAD and LUSC was assessed. A discriminant analysis combining these three miRNAs accurately discriminated between LUAD and LUSC in both the test and validation samples, with sensitivities and specificities of 76% and 80%, and 85% and 83%, respectively. They can be identified as biomarkers capable of distinguishing between LUAD and LUSC in the lungs [77]. Another study developed an assay based on *miR-21*, *miR-205*, and *miR-375*. This method accurately identified the LUAD/LUSC histotype in 25 biopsies with 96% accuracy and correctly classified all 12 cases where histopathological examination of the biopsies was incorrect. Furthermore, examination of publicly available datasets revealed *miR-205* and *miR-375* as the most reliable miRNAs for tissue typing of LUAD and LUSC, and the levels of these two miRNAs were not affected by tumor pathological stage, age, or race [78]. In another study, a logistic model was generated using seven candidate miRNAs (*miR-29a*, *miR-29b*, *miR-34a*, *miR-375*, *miR-205*, *miR-25*, and *miR-27a*) to discriminate between LUSC and LUAD. This model demonstrated a balanced accuracy of 96.0%. The miRNA panel was validated in an independent cohort of 68 FFPE surgical specimens, achieving a balanced accuracy of 97.6% and an area under the curve (AUC) value of 0.982. These results confirm the high diagnostic accuracy of the miRNA panel in distinguishing LUSC from LUAD in surgical specimens. Additionally, for LUSC and LUAD patients, the cytological diagnosis rate (81.2–71.8% and 29.0–55.0% for LUAD and LUSC, respectively) was significantly lower than that of the miRNA panel (91.8% and 88.4% for LUAD and LUSC, respectively) [79].

Furthermore, pri-miRNAs have emerged as a promising group of potential cancer biomarkers due to their high diagnostic accuracy for tumor detection. The expression of *miRNA-3662* and its precursor (*pri-miRNA-3662*) was analyzed in 56 fresh-frozen NSCLC tissues and corresponding adjacent non-cancerous tissues. The study revealed significant overexpression of *miRNA-3662* and *pri-miRNA-3662* in LUAD compared to LUSC and adjacent non-cancerous tissues. Combined analysis of *pri-miRNA-3662* and mature *miRNA-3662* enabled differentiation of LUAD tissue from LUSC with a sensitivity of 96% and a specificity of 85.7% [80].

The presence of ground glass nodules (GGN) in the lungs has persistently posed a problem. Although GGN is strongly suggestive of lung cancer, specifically LUAD, it may also indicate a completely benign process. Early identification of GGN patients at high risk for LUAD is particularly crucial for effective treatment and improved prognosis. A 7-miRNA panel was developed to effectively analyze whether GGNs in patients are benign or indicative of LUAD. The 7-miRNA panel demonstrated a sensitivity of 86.4% and specificity of 60.6% in detecting LUAD [81].

### 4.2. Extracellular Fluid

Altered expression of miRNAs is well recognized to contribute to cancer development and invasion through post-transcriptional gene silencing. Numerous studies have demonstrated that changes in circulating miRNAs are associated with LUAD, indicating their potential for non-invasive detection of the disease. Recently, the analysis of miRNAs in easily accessible body fluid sources such as serum, plasma, whole blood, and sputum has gained significant interest. Detecting miRNAs in body fluids can facilitate the early identification of LUAD patients. The differential expression patterns of these miRNAs can be detected at various stages, ranging from early to progressive stages and even after cancer metastasis, enabling real-time and dynamic monitoring of changes. Therefore, there is an urgent need to identify minimally invasive biomarkers for early diagnosis [82].

#### 4.2.1. Blood

Interest in circulating RNAs is rapidly increasing as their potential as biomarkers is recognized. Blood test is the most commonly used and convenient test in clinic. In 2012, the global expression of miRNAs in whole blood was studied to differentiate LUAD from controls. The possibility of miRNAs as biomarkers for the diagnosis of lung cancer was explored using two different methods with accuracy, sensitivity and specificity values ranging from 86% to 100%. *MiR-190b*, *miR-630*, *miR-942*, and *miR-1284* emerged as the principal miRNAs identified in this experimental investigation [83]. Subsequent study isolated RNA from 80 serum samples and identified six miRNAs at significantly higher levels and two miRNAs at significantly lower levels in LUAD serum. Differences in miRNA profiles were further demonstrated to support the potential of circulating miRNAs as diagnostic biomarkers for LUAD [84]. Furthermore, Mei Chee Tai developed a serum miRNA-based diagnostic classifier by conducting an in-depth bioinformatics analysis of miRNA profiles in a training cohort consisting of 143 LUAD patients and 49 healthy subjects. This classifier was then validated on an independent sample cohort, comprising of LUAD patients, healthy subjects, and patients with benign lung disease, and showed a sensitivity of 89.1%, a specificity of 94.9%, and an AUC value of 0.958. Notably, the classifier correctly identified 90.8% of stage I LUAD cases [85]. Gao investigated the miRNA profiles in individuals with aggressive stage I LUAD. The study included a total of 460 participants, consisting of 254 LUAD patients, 76 patients with benign pulmonary nodules (BPNs), and 130 healthy controls (HCs). They developed a diagnostic signature (d-signature) comprising four extracellular vesicle (EV)-derived miRNAs (*miR-450b-5p*, *miR-3615*, *miR-106b-3p*, and *miR-125a-5p*) for the early detection of LUAD. The d-signature demonstrated high accuracy, with area under the curve (AUC) values of 0.917 and 0.902 in the training and test cohorts, respectively. Importantly, the d-signature could distinguish adenocarcinoma in situ (AIS) and minimally invasive adenocarcinoma (MIA) patients, achieving AUC values of 0.846 and 0.92, respectively [86].

In addition to miRNA profiles, individual miRNAs can also constitute diagnostic markers for early detection of LUAD. The expression of *miR-155* in the serum of LUAD patients was significantly higher than that of the normal control group. The detection of serum *miR-155* levels showed much higher sensitivity than CA-125 or CEA. In addition, when combined with CA-125 detection, *miR-155* obtained competitive sensitivity and specificity in the diagnosis of LUAD. Endogenous *miR-155* is stably present in patient serum, allowing for sensitive and specific detection [87]. It is critical to accurately assess the operability of LUAD for effective patient management. Circulating *miR-3662* in plasma demonstrated a strong correlation with the operability of LUAD. Moreover, a higher stage of lung cancer was found to be associated with increased miRNA expression [88]. This study shows that miRNAs also hold promise in determining the operability of LUAD.

#### 4.2.2. Phlegm

MiRNA has been found to exist stably in sputum, and its detection in sputum has shown promise for diagnosing LUAD. A combination of *miR-21*, *miR-486*, *miR-375*, and *miR-200b* has been identified to differentiate between LUAD patients and healthy individuals, with a sensitivity of 80.6% and specificity of 91.7%. The marker panel has also been validated in an independent population, demonstrating improved sensitivity and specificity compared to using any single marker alone [89].

#### 4.2.3. Pleural Fluid

The expression levels of circulating extracellular miRNAs in patients with pleural effusion may be useful for diagnosing LUAD-associated malignant pleural effusion (LA-MPE) and distinguishing it from benign pleural effusion (BPE). The same research group performed two studies on pleural effusion patients. First, they used quantitative PCR to detect the differences in the levels of *miR-134*, *miR-185*, and *miR-22*, which were all shown to be significantly downregulated in 45 LA-MPE patients compared to 42 BPE patients. The AUC values for *miR-134*, *miR-185*, *miR-22*, and CEA (carcinoembryonic antigen), common tumor marker, were 0.721, 0.882, 0.832, and 0.898, respectively. Combining CEA with the three miRNAs improved diagnostic performance, resulting in an AUC of 0.942, a sensitivity of 91.9%, and a specificity of 92.5% [90]. In the following study, they used microarray to evaluate expression of 160 miRNAs in two matched groups (*n* = 10) with MPE and BPE, which were then validated in the patient group from the first study. Here, *miR-198* was found to be significantly downregulated in LA-MPE. In the validation set, *miR-198* and CYFRA 21-1 exhibited AUC values of 0.887 and 0.836, respectively. The diagnostic ability of miR-198 is comparable to or even better than that of CEA and CYFRA 21-1, both of which are classic tumor markers commonly used in the clinic. The combined AUC for all three markers was 0.926, with a sensitivity of 89.2% and a specificity of 85.0%, suggesting that the new method was not an improvement over the first one [91].

#### 4.2.4. Exosome

Exosomes are small secretory vesicles with a diameter of 20–150 nm, which are small spherical microvesicles produced by the exocytosis of multivesicular bodies. Exosomes can be secreted by cells under almost any physiological and pathological conditions and can also be found in body fluids such as serum, urine, saliva, and amniotic fluid. Exosomes contain various components such as DNA, lipids, proteins, miRNA, single-stranded RNA, and lncRNA, among others. These components can be transferred to recipient cells, mediating diverse biological processes including metastasis, tumorigenesis, and immune response [92]. The role of exosome-secreted miRNAs is being extensively studied, and they may serve as diagnostic biomarkers for LUAD.

A study conducted a circulating exosomal miRNA analysis on 46 stage I NSCLC patients and 42 healthy individuals. The findings indicate that *miR-181-5p*, *miR-30a-3p*, *miR-30e-3p*, and *miR-361-5p* are specifically associated with LUAD, while *miR-10b-5p*, *miR-15b-5p*, *and miR-320b* are specific to LUSC. These miRNAs show promise as highly sensitive, non-invasive biomarkers for early diagnosis [93]. In another study, elevated levels of circulating exosomal *miR-342-5p*, *miR-574-5p*, and *miR-222-3p* were observed in LUAD patients compared to healthy controls. However, their expression levels significantly decreased after tumor resection when analyzing preoperative and postoperative samples. Additionally, *miR-342-5p* and *miR-574-5p* exhibited increased expression in LUAD tissues compared to para-cancerous tissues, while *miR-222-3p* did not. The combined evaluation of *miR-342-5p* and *miR-574-5p* demonstrated a promising diagnostic potential with an AUC of 0.813, sensitivity of 80.0%, and specificity of 73.2% [94]. 

## 5. MiRNAs as Prognostic Biomarkers in LUAD

Studying miRNA expression profiles in different tissues may provide efficient and reliable biomarkers to predict disease outcome. As early as 2006, high *miR-155* and low *let-7a-2* expression were associated with poorer survival in both univariate and multivariate analyses [95]. In the Maryland cohort, elevated *miR-21* was associated with worse LUAD-specific mortality. When evaluated in two other cohorts, *miR-21* was also associated with worse LUAD-specific mortality in the Norwegian cohort and worse recurrence-free survival in the Japanese cohort. More advanced LUAD patients expressed significantly higher levels of *miR-21* compared with TNM stage I tumors [96]. Normal lung tissues from all 23 cases with a pathological diagnosis of LUAD in the tissue bank were matched. In 19 (83%) cases, *miR-29b* was downregulated in tumors compared with matched normal lung tissue. Analysis using the median tumor level found that *miR-29b* expression levels were highly correlated with overall survival (OS) and event-free survival (EFS) [97]. The expression levels of *miR-381* and *miR-708* were also reported to be significantly correlated with EFS and OS, respectively [56,98]. Abnormal methylation of miRNAs can also affect the prognosis of LUAD. Patients with increased *miR-34b/c* methylation had significantly shorter EFS and OS compared to those with no methylation or low levels of *miR-34b/c* methylation. Epigenetic inactivation of *miR-34b/c* by DNA methylation has independent prognostic value in early-stage LUAD patients [99]. Furthermore, *miR-214* expression in LUAD was found to be associated with bone metastasis. High expression of *miR-214* in LUAD promoted osteoblast differentiation and facilitated intercellular communication between osteoblasts and osteoclasts through exosomal miRNAs. This disruption of bone homeostasis enhanced bone resorption, favoring cancer cell migration, proliferation, and colonization, ultimately leading to bone metastasis. Circulating exosomal *miR-214* levels showed potential for predicting the risk of bone metastasis [100]. In addition to the aforementioned miRNAs, we have also compiled other miRNAs that are associated with the prognosis of LUAD in Table 5.

In recent years, an increasing number of studies have reported the involvement of miRNAs in cancer metastasis, including brain metastasis (BM) and lymph node metastasis, which are common complications of LUAD. The incidence of locally advanced LUAD combined with BM can be as high as 30–50%. Zhao et al. screened important brain metastasis-associated miRNAs from 77 LUAD patients with brain metastases (BM+) or without brain metastases (BM-). Predictive models were developed using the random forest supervised classification algorithm and the class center method. These models were trained on a set of 42 patients and then validated on a separate test set of 35 patients. A predictive model including *miR-210*, *miR-214*, and *miR-15a* was created to classify patients into two groups with significantly different subtypes of brain metastases with 90.4% accuracy. The similar predictive power of 91.4% accuracy was observed in the test cohort [101]. Furthermore, the expression of *miR-423-5p* was significantly increased in LUAD cases with brain metastasis compared to those without. A combination of imaging, histological, and molecular analyses revealed that overexpression of *miR-423-5p* significantly enhanced local invasion, distant brain metastasis, and tumor burden. Expression levels of MTSS1 were inversely correlated with *miR-423-5p* upregulation in LUAD specimens and showed a correlation with survival in brain metastasis patients [102]. In another study, *miR-31* was found to be upregulated in node-positive patients in a separate cohort. The role of *miR-31* as a marker of lymph node metastasis was validated in 233 LUAD cases at The Cancer Genome Atlas (TCGA). Furthermore, *miR-31* was found to enhance cell migration, invasion, and proliferation through an ERK1/2 signaling-dependent mechanism. Notably, *miR-31* emerged as a significant predictor of survival in multivariate Cox regression models, even when controlling for cancer stage. Additionally, reduced expression of *miR-31* was associated with a favorable prognosis in patients with T2N0 stage cancer [103]. These findings indicate that miRNA can serve as an independent predictor strongly associated with brain metastasis and lymph node metastasis, offering high sensitivity and specificity in clinical practice.

## 6. MiRNAs in LUAD Therapy

MiRNAs hold tremendous potential in cancer therapy owing to their multifaceted biological roles. They can simultaneously suppress multiple gene targets involved in LUAD development, and even a small amount of miRNA can reverse the malignant phenotype. However, the utilization of miRNAs in LUAD therapy poses several challenges, including targeted delivery, uptake by cancer cells, and safety considerations. Table 6 provides a selection of miRNAs that could be utilized for LUAD treatment, while Appendix A encompasses a comprehensive list of miRNAs associated with LUAD therapy.

### 6.1. MiRNAs as Potential Drugs for LUAD Treatment

As discussed in Section 2, miRNAs play a dual role as oncogenes and tumor suppressors in LUAD and thus have potential as cancer therapeutics. Synthetic miRNA mimics or antagomirs have emerged as a promising strategy for treating lung cancer. MiRNA mimics and anti-miRNAs have been successfully employed to restore normal gene networks in tumor cell lines, xenograft models, and clinical trials [104]. To restore downregulated miRNA levels (tumor suppressors), miRNA mimics or miRNA expression vectors can be synthesized, while chemically modified antisense nucleotides (anti-miRNAs) are commonly used to decrease the abundance of upregulated miRNAs (oncomiRs). In lung cancer, *let-7*, *miR-34*, *miR-126*, *miR-200c*, *miR-145*, and *miR-150* have undergone extensive investigation as potential miRNA-based therapies. The *let-7* miRNA was the first to be explored for miRNA replacement therapy in lung cancer. A549 cells, which carry *KRAS* mutations, were transiently transfected with synthetic *let-7* and subsequently subcutaneously transplanted into NOD/SCID mice. This led to a delayed tumor growth in the group transfected with synthetic *let-7*. The antitumor effect of *let-7* was further examined by intranasally administering a *let-7*-expressing adenovirus (Ad.*let-7*) in a mouse LUAD model, resulting in a significant reduction in tumor growth among the transfected mice [105]. Carla et al. developed aptamer-miRNA conjugates as multifunctional molecules that hinder the growth of Axl-expressing tumors using aptamers that bind to and antagonize the oncogenic receptor tyrosine kinase Axl (GL21.T). By combining *let-7g* with GL21.T, they demonstrated selective delivery to target cells, processing by the RNA interference machinery, and silencing of *let-7g* target genes. Notably, this multifunctional conjugate reduced tumor growth in a LUAD xenograft model [106]. In another study, chondroitin sulfate (CS) was chemically modified via Michael addition onto polyamidoamine dendrimers (PAMAM) to construct a tumor-targeting vector called CS-PAMAM, which efficiently delivered *miR-34a*. Intravenous administration of the CS-PAMAM/*miR-34a* complex effectively suppressed tumor growth and induced tumor cell apoptosis in mice bearing A549 xenografts, attributed to the increased accumulation of *miR-34a* in tumor tissues [107]. Furthermore, Lv et al. synergistically utilized *miR-126-3p* and biomimetic nanoparticles to achieve efficient transduction of miRNA into lung cancer cells, demonstrating a promising approach for effective therapy against LUAD [108]. These findings suggest that miRNA replacement therapy holds promise for treating LUAD. Considering the significant regulatory role of miRNAs in cancer initiation and progression, numerous preclinical studies have demonstrated the tremendous potential of miRNAs in cancer therapy. Several pharmaceutical companies have initiated research and development in miRNA therapeutics for cancer treatment [109]. Other than directly targeting cancer processes, miRNAs may also be utilized to supplement conventional cancer therapies.

### 6.2. MiRNA and Chemo-Resistance

Chemotherapy is the primary treatment approach for lung cancer, offering improved survival and quality of life, particularly in advanced cases. However, multidrug resistance (MDR) remains a significant challenge in successful chemotherapy for cancer patients. Recent studies have revealed the pivotal involvement of miRNAs in chemotherapy-induced drug resistance.

Cisplatin, a conventional chemotherapy drug for LUAD, faces hurdles due to cisplatin resistance in clinical applications. Glutathione S-transferase P1 (GSTP1) is known to contribute to cisplatin resistance. A549/cisplatin (CDDP) cells exhibited a 2.7 ± 0.38 fold upregulation of GSTP1 mRNA expression compared to parental A549 cells, while *miR-513a-3p* expression was downregulated by 0.34 ± 0.03-fold. *MiR-513a-3p* was shown to sensitize human LUAD cells to cisplatin by targeting GSTP1 [110]. Another downregulated miRNA in A549/CDDP, *miR-181b*, when overexpressed, enhanced sensitivity to anticancer drugs by reducing BCL2 protein levels and promoting CDDP-induced apoptosis [111]. *MiR-27a* was found to regulate EMT and cisplatin resistance in vitro, as well as the response of LUAD cells to cisplatin in vivo. Higher *miR-27a* expression was detected in tumor tissue samples from LUAD patients receiving cisplatin-based chemotherapy, correlating with lower RKIP expression, decreased cisplatin sensitivity, and poor prognosis [112].

MiRNAs have also been identified as regulators of resistance to other chemotherapeutic drugs. In docetaxel-resistant SPC-A1/DTX cells, *miR-100* was significantly downregulated compared to parental SPC-A1 cells. Ectopic *miR-100* expression resensitized SPC-A1/DTX cells to docetaxel by inhibiting cell proliferation, inducing G2/M phase cell arrest, and promoting apoptosis [113]. Upregulation of *miR-200b* significantly improved the response of SPC-A1/DTX cells to docetaxel in a nude mouse xenograft model. A luciferase reporter gene was employed to demonstrate that *miR-200b* can directly target E2F3. Reduced expression of *miR-200b* was also observed in tumor tissues from LUAD patients undergoing docetaxel-based chemotherapy and was associated with high E2F3 expression, reduced docetaxel sensitivity, and a poor prognosis [114].

Furthermore, miRNAs can serve as predictive markers for chemotherapy response in patients. Data from Shi’s study suggest *miR-25*, *miR-145*, and *miR-210* as potential predictors of response to pemetrexed maintenance therapy in patients with EGFR-mutated or ALK translocation-negative LUAD [115].

These studies underscore the critical role of miRNAs in drug sensitivity and chemotherapy resistance. MiRNAs offer a promising avenue for overcoming LUAD chemotherapy resistance in the future.

### 6.3. MiRNA and Radiation Therapy

Radiation therapy is a widely used treatment for LUAD, and multiple studies have demonstrated the potential of miRNAs in regulating the radiosensitivity of lung cancer cells, thereby enhancing the efficacy of radiotherapy.

In LUAD cells, *miR-195-5p* exhibits downregulation, while its overexpression inhibits cell proliferation and invasion while augmenting cell sensitivity to radiotherapy. *MiR-195-5p* achieves this effect by specifically downregulating the HOXA10 gene [116]. *MiR-511* facilitates the expression and activation of the BAX protein through TRIB2, thereby increasing the sensitivity of LUAD cells to radiotherapy [117]. Potassium voltage-gated channel subfamily Q member 1 opposite strand 1 (KCNQ1OT1) induces autophagy by acting as a sponge for *miR-372-3p*, thereby contributing to SBRT (stereotactic body radiotherapy) resistance in LUAD. Targeting KCNQ1OT1 represents a potential strategy to enhance the antitumor effect of radiotherapy in LUAD [118]. Radiation therapy reduces the expression of the VANGL1 gene in LUAD cells. *MiR-29b-3p* can upregulate the expression of the VANGL1 gene, which enhances the tolerance of LUAD cells to radiation therapy. This results in a decrease in apoptosis and DNA double-strand breaks after treatment and mitigates the adverse effects of radiotherapy on LUAD [119]. Both in vitro and in vivo experiments have demonstrated that elevated *miR-126-5p* inhibits cell migration, promotes apoptosis, and enhances the sensitivity of LUAD cells to radiation therapy through the EZH2/KLF2/BIRC5 axis. *MiR-126-5p* downregulates EZH2 to sensitize LUAD cells to radiation therapy targeting KLF2/BIRC5 [120]. Overall, miRNAs play a critical role in regulating the radiosensitivity of lung cancer cells, offering a promising strategy to enhance the effectiveness of radiotherapy in treating LUAD.

### 6.4. MiRNA and EGFR

Over the past decade, extensive therapeutic research on LUAD has primarily focused on the EGFR pathway and genetic variations in EGFR. EGFR is one of the most prevalent proto-oncogenes in LUAD, with nearly all specific EGFR mutations involving a leucine-to-arginine change at position 858 and a deletion in exon 19 [121]. The roles of miRNAs in the EGFR signaling network have gradually emerged. Both cell and in vivo experiments demonstrated that *miR-145* had a significant inhibitory effect on LUAD cell proliferation by directly targeting EGFR [38]. The expression level of *miR-138-5p* is markedly reduced in gefitinib-resistant LUAD cells, and *miR-138-5p* can reverse the response of LUAD cells to gefitinib by targeting the negative regulatory G protein-coupled receptor 124 (GPCR124) [122]. In EGFR-TKI-resistant LUAD cells, *miR-7* exhibits significantly lower expression compared to sensitive cells. Liposome-mediated delivery of the *miR-7* expression plasmid can reverse the resistance of LUAD cells to EGFR-TKI by targeting multiple key genes, including EGFR, PIK3CD, IRS1, and BCL2L1 [123]. Furthermore, *miR-608* and *miR-4513* significantly enhances the antiproliferative effect of gefitinib in LUAD cells. Patients with high expression levels of *miR-608* and *miR-4513* exhibit better efficacy in response to EGFR-TKI treatment. The expression levels of *miR-608* and *miR-4513* hold potential for predicting the response and prognosis of LUAD patients to EGFR-TKI treatment [124]. The study of miRNAs and polymorphisms offers clinical potential for personalized treatment decisions and holds great promise in LUAD research.

## 7. MiRNA and Tumor Immunity in LUAD

The immune system has the inherent ability to recognize and eliminate tumor cells within the tumor microenvironment. However, tumor cells can employ various mechanisms to suppress the immune system, enabling their survival and evading the antitumor immune response. These characteristics of tumor cells are commonly referred to as immune escape. Tumor immunotherapy aims to control and eliminate tumors by reactivating and sustaining the tumor-immune cycle, thus restoring the body’s natural antitumor immune response. This therapeutic approach encompasses immune checkpoint inhibitors, therapeutic antibodies, cancer vaccines, cell therapy, and small molecule inhibitors.

Among these approaches, monoclonal antibodies targeting immune checkpoint proteins have been extensively studied and represent a significant advancement in clinical trials involving patients with NSCLC. The checkpoints under investigation include cytotoxic T lymphocyte-associated antigen-4 (CTLA-4), programmed death-1 (PD-1), and its ligands PD-L1 (B7H1) and PD-L2 (B7-DC). These checkpoints respectively regulate early and late T cell activity within the tumor microenvironment [125]. Studies have identified certain miRNAs that play a role in the tumor immune response process. For instance, *miR-33a* has been found to exhibit a negative correlation with the expression of PD-1, PD-L1, and CTLA4. Patients with high *miR-33a* expression demonstrated significantly better prognosis in LUAD. Analysis of the TCGA database also revealed that elevated levels of *miR-33a* were associated with lower PD-1 expression and extended survival in a larger population [126]. In the LUAD model, *p53* was found to regulate PDL1 through *miR-34*, which directly binds to the 3′ untranslated region of PDL1 [127]. Furthermore, targeting the *miR-30-5p* family to inhibit USP22 prevents the induction of PD-L1 expression under hypoxic conditions, thereby impeding the ability of activated T cells to kill LUAD cells [128].

In the absence of external intervention, the number of T cells capable of recognizing tumor cells within the human body is limited. Cell therapy, also known as adoptive immunotherapy or adoptive T cell transfer (ACT), aims to externally modify ordinary immune cells, enabling them to recognize tumors and trigger an immune response against tumor cells. In NK cells from LUAD patients, *miR-218-5p* was found to be upregulated, while SHMT1 was downregulated. Stimulation with interleukin 2 (IL-2) reversed this expression pattern. The addition of *miR-218-5p* reduced IL-2-induced cytokine expression and cytotoxicity against LUAD cells in NK-92 cells. Additionally, *miR-218-5p* negatively regulated SHMT1, attenuating the effects of *miR-218-5p* on cytotoxicity, IFN-γ, and TNF-α secretion in IL-2-activated NK cells. Depletion of *miR-218-5p* promoted NK cell killing and suppressed tumor growth [129]. In LUAD patients, *miR-582* expression levels were increased, while CD1B expression was decreased. Moreover, *miR-582* can directly inhibit the function of dendritic cells by targeting the CD1B gene. Downregulation of CD1B and upregulation of *miR-582* are correlated with poor prognosis in patients [130].

MiRNAs also play a role in regulating immune cell infiltration, a crucial step in the immune response where immune cells invade tissues. The expression of *miR-125b-5p* and *miRNA-30a-5p* is associated with immune cell infiltration in LUAD [131,132]. The SNX20AR/*miRNA-301a-3p*-mediated reduction of SNX20 is linked to lung cancer progression and immune infiltration in LUAD [133]. Therefore, miRNAs hold significant potential in the field of tumor cell immunotherapy.

## 8. Conclusions and Future Prospects

MiRNAs have emerged as a promising tool for cancer therapy due to their ability to regulate multiple biological pathways. Dysregulation of numerous miRNAs has been observed in various types of cancer, and even subtle changes in their levels can significantly impact disease outcomes. In the context of LUAD, miRNAs act as potent inhibitors of gene expression, effectively interfering with cancer cell growth and survival. Furthermore, miRNAs exhibit greater stability in serum, plasma, and FFPE-preserved samples compared to mRNAs, making them ideal non-invasive biomarkers for monitoring disease progression and classifying cancer subtypes. Increasing evidence suggests that miRNAs have the potential to combat chemotherapy-induced drug resistance. However, it is important to consider that miRNA-based therapies may entail unpredictable side effects, as each miRNA can target hundreds of mRNAs, even beyond the intended specific target mRNA. In accordance with this, only a handful of miRNAs have entered clinical trials and almost all have been abandoned before they reached phase 3. In addition, in preclinical trials, some miRNAs have been found to be both tumor suppressors and oncogenes, demonstrating the importance of the expression landscape into which they are introduced. Identifying downstream elements to which miRNAs involved in cancer converge and targeting them with more specifically designed siRNAs might be a more promising strategy. Moreover, miRNAs derived from exosomes hold great promise in the diagnosis, prognosis, and treatment of LUAD. Further research on exosomal miRNAs in LUAD is expected to enhance our understanding of this previously overlooked miRNA fraction.

To effectively translate these foundational research findings into clinical practice, a comprehensive understanding of miRNA biology is crucial. Researchers have focused on identifying miRNA signatures that may offer new insights into longstanding questions. However, ensuring the safety and efficacy of miRNA-based therapies necessitates targeted delivery to tumor sites, efficient uptake by cancer cells, and minimizing off-target effects. Establishing standardized methods for miRNA detection, improving our understanding of the interactions between miRNAs and other genomic elements, and developing biocompatible delivery vehicles for miRNAs to target lung lesions are paramount.

In conclusion, there are still obstacles to overcome on the path to translating miRNA research into clinical practice. However, with persistent efforts and emerging research findings, we can overcome these challenges and pave the way for a new era of comprehensive miRNA application in LUAD in the near future.

## Figures and Tables

**Figure 1 ijms-24-13302-f001:**
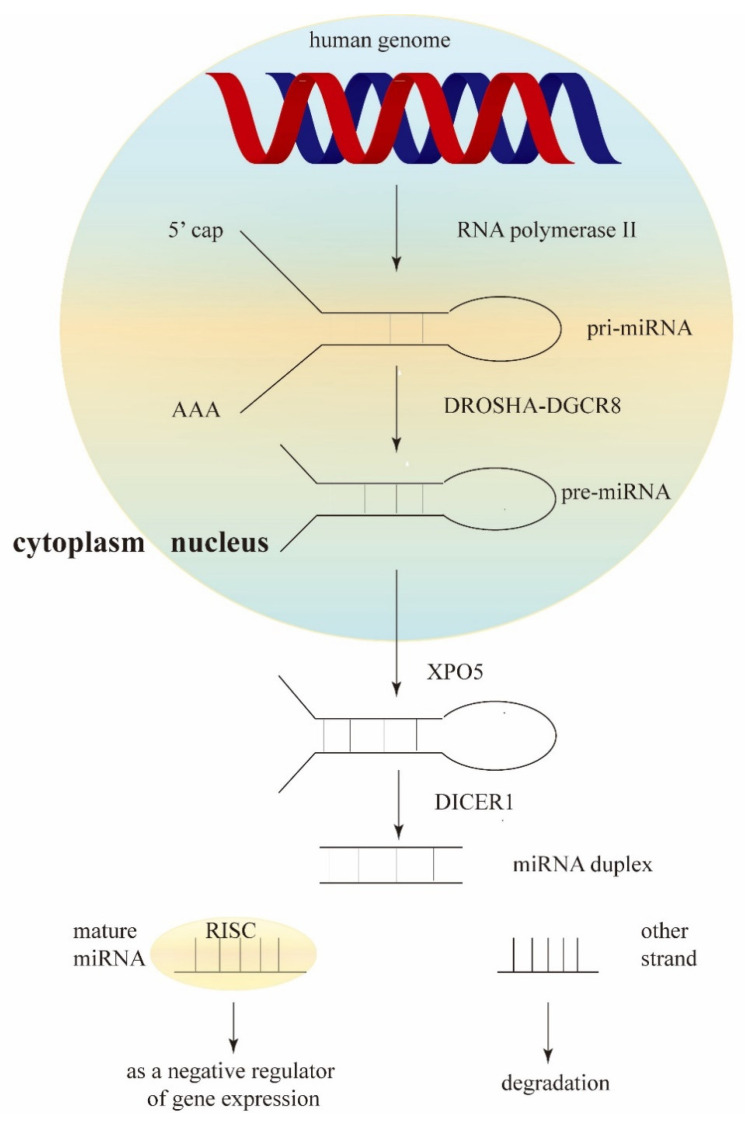
Schematic representation of miRNA biogenesis.

**Table 1 ijms-24-13302-t001:** Tumor suppressor miRNAs in LUAD.

MiRNA	Possible Target	Identified Effects	References (PMID and Link accessed on 22 August 2023)
miR-1	FAM83A	suppress A549 cell growth and motility	33266425 (https://www.mdpi.com/1422-0067/21/22/8833)
miR-7	BCL-2	inhibit A549 cell proliferation, migration and induce apoptosis	21750649 (https://www.ijbs.com/v07p0805.htm)
miR-22	ErbB3	exhibit excellent anticancer activity both in vitro and in vivo	22484852 (https://link.springer.com/article/10.1007/s00432-012-1194-2)
miR-23b	cyclin D1	inhibit the proliferation and migration	28976503 (https://pubs.rsc.org/en/content/articlelanding/2017/BM/C7BM00599G)
miR-98	TGFBR1	inhibit proliferation and metastasis in A549 cell	30387848 (https://www.spandidos-publications.com/ijo/54/1/128)
miR-125a	STAT3	inhibit the proliferation, invasion and metastasis	31930562 (https://onlinelibrary.wiley.com/doi/10.1002/jcb.29586)
miR-126	ADAM9	inhibit lung adenocarcinoma (LUAD) development and progression	36171576 (https://molecular-cancer.biomedcentral.com/articles/10.1186/s12943-022-01651-4)
miR-142	NR2F6	suppress the proliferation, migration and invasion	31168689 (https://link.springer.com/article/10.1007/s13577-019-00258-0)
miR-144	EZH2	contribute to progression of LUAD	30280514 (https://onlinelibrary.wiley.com/doi/10.1002/cam4.1714)
miR-145	EGFR/NUDT1	inhibit cell proliferation of human LUAD	21289483 (https://www.tandfonline.com/doi/abs/10.4161/rna.8.1.14259)
miR-149	RAP1B	inhibit the progression of LUAD	32432747 (https://www.europeanreview.org/article/21173)
miR-150	TNS4	inhibit LUAD cell malignancy	31052206 (https://www.mdpi.com/2072-6694/11/5/601)

**Table 2 ijms-24-13302-t002:** Oncogenic miRNAs in LUAD.

MiRNA	Possible Target	Identified Effects	References (PMID and Link accessed on 22 August 2023)
miR-9	ID4	promotes LUAD cell progression	34723712 (https://journals.sagepub.com/doi/full/10.1177/15330338211048592)
miR-10b	KLF4	promotes A549 cell proliferation and invasion	24216130 (https://eurjmedres.biomedcentral.com/articles/10.1186/2047-783X-18-41)
miR-19	PTEN	triggers EMT of LUAD cells accompanied by growth inhibition	26098000 (https://www.laboratoryinvestigation.org/article/S0023-6837(22)01359-9/fulltext)
miR-21	SET/TAF-Iα	promotes LUAD progression	31176779 (https://www.sciencedirect.com/science/article/abs/pii/S002432051930459X)
miR-93	PTEN, RB1	plays an oncogenic role by inhibiting PTEN and RB1	29309884 (https://www.sciencedirect.com/science/article/abs/pii/S0378111918300313)
miR-96	ARHGAP6	an oncogene in LUAD and facilitate tumor progression	34338998 (https://link.springer.com/article/10.1007/s13353-021-00652-1)
miR-183	PECAM1	positive influence on LUAD cell viability and proliferation	29749535 (https://www.spandidos-publications.com/or/40/1/83)
miR-196a	ANXA1	promotes migration and invasion	33775710 (https://www.sciencedirect.com/science/article/abs/pii/S0304383521001324)
miR-196b	RSPO2	promotes proliferation, migration and invasion	33402849 (https://www.ncbi.nlm.nih.gov/pmc/articles/PMC7778444/)

**Table 3 ijms-24-13302-t003:** MiRNAs involved in ceRNA networks in LUAD.

MiRNA	Possible Target	Identified Effects	References (PMID and Link accessed on 22 August 2023)
miR-7	IRS2	circFAT1 promotes tumorigenesis through sequestering miR-7	35844799 (https://www.ijbs.com/v18p3944.htm)
miR-9	CPEB3	linc00968/miR-9/CPEB3 regulatory axis plays a critical role in LUAD	33159015 (https://www.aging-us.com/article/103833)
miR-17	QKI-5	circ-MTO1/miR-17/QKI-5 feedback loop inhibits LUAD progression	30975029 (https://www.tandfonline.com/doi/full/10.1080/15384047.2019.1598762)
miR-18b	VMA21	lncRNA ZFPM2-AS1 promotes proliferation via miR-18b/VMA21 axis in LUAD	31297866 (https://onlinelibrary.wiley.com/doi/10.1002/jcb.29176)
miR-20a	SLC7A5	circRNA LDLRAD3 enhances the malignant behaviors of LUAD cells via the miR-20a-5p-SLC7A5 axis	35035814 (https://www.hindawi.com/journals/jhe/2022/2373580/)
miR-20b	CCND1	linc00467 promotes LUAD proliferation via sponging miR-20b-5p	31686834 (https://www.ncbi.nlm.nih.gov/pmc/articles/PMC6709798/)
miR-22	BCL2	lncRNA DGCR5 promotes LUAD progression via inhibiting miR-22	29030962 (https://onlinelibrary.wiley.com/doi/10.1002/jcp.26215)
miR-26a	E2F7	SNHG6 may act as an oncogenic lncRNA in LUAD carcinogenesis by regulating the miR-26a-5p/E2F7 axis	30257360 (https://www.sciencedirect.com/science/article/abs/pii/S0753332218341921)
miR-29b	STAT3	lncRNA H19 promotes viability and EMT of LUAD cells by targeting miR-29b-3p and modifying STAT3	30747209 (https://www.spandidos-publications.com/ijo/54/3/929)
miR-33b	GPAM	lncRNA MSC-AS1 facilitates LUAD through sponging miR-33b-5p to upregulate GPAM	33821667 (https://cdnsciencepub.com/doi/full/10.1139/bcb-2020-0239)
miR-34	PDL1	has-circRNA-002178 could enhance PDL1 expression via sponging miR-34 in LUAD cells to induce T-cell exhaustion	31949130 (https://www.nature.com/articles/s41419-020-2230-9)
miR-96	CYLD	lncRNA GMDS-AS1 inhibits LUAD development by regulating miR-96-5p/CYLD signaling	31860169 (https://onlinelibrary.wiley.com/doi/10.1002/cam4.2776)
miR-98	AKR1B10-ERK	linc00665 promotes LUAD progression and functions as ceRNA to regulate AKR1B10-ERK signaling by sponging miR-98	30692511 (https://www.nature.com/articles/s41419-019-1361-3)
miR-100	SMARCA5	lncRNA HAGLROS facilitates the malignant phenotypes via repressing miR-100 and upregulating SMARCA5	35307327 (https://www.sciencedirect.com/science/article/pii/S2319417020302365)

**Table 4 ijms-24-13302-t004:** MiRNAs associated with LUAD diagnosis.

MiRNA	Sample Source and Identification Effects	Statistics	Patient Stage, *n* (%)	References (PMID and Link accessed on 22 August 2023)
miR-10b	MiR-10b in extracellular vesicles may be a potential diagnostic biomarker for LUAD	AUC = 0.998, sensitivity = 98.75%, specificity = 98.55%	I 58 (72.5%) II 16 (20.0%) III 5 (6.3%) Unknown 1(1.2%)	34257722 (https://www.spandidos-publications.com/10.3892/ol.2021.12875)
miR-126	Bronchoalveolar lavage fluid exosomal miR-126 could serve as diagnostic biomarkers in early-stage LUAD	/	IA 8 (61.5%), IB 2 (15.4%), IIA 3 (23.1%)	29806739 (https://onlinelibrary.wiley.com/doi/10.1111/1759-7714.12756)
miR-130a	MiR-130A as a diagnostic marker to differentiate malignant mesothelioma from LUAD in pleural effusion cytology	AUC = 0.70, sensitivity = 77%, specificity = 67%	/	28449331 (https://acsjournals.onlinelibrary.wiley.com/doi/10.1002/cncy.21869)
miR-505	Extracellular vesicle-delivered miR-505-5p as a diagnostic biomarker for early-stage LUAD	AUC = 0.899, sensitivity = 83.3%, specificity = 93.3%	/	30864684 (https://www.spandidos-publications.com/ijo/54/5/1821)
miR-19b, miR-183	Plasma-derived miR-19b, miR-183 can be used to identify lung cancer and miR-183 was more effective in discriminating LUAD from healthy individuals	AUC = 0.990, sensitivity = 94.7%, specificity = 95.2%	I - II 24 (32.0%) III 47 (62.7%) IV 4 (5.3%)	27768748 (https://journals.plos.org/plosone/article?id=10.1371/journal.pone.0165261)
miR-339, miR-21	Plasma-based miR-339 and miR-21 evaluation can serve as the tumor markers for LUAD screening	AUC = 0.963, sensitivity = 92.9%, specificity = 92.9%	IA 12 (42.9%), IB 3 (10.7%), IIA 2 (7.1%), IIB 5 (17.9%), IIIA 4 (14.3%), IIIB 2 (7.1%)	29103767 (https://www.sciencedirect.com/science/article/abs/pii/S0344033817308737)
miR-4529, miR-8075, miR-7704	Three miRNA integrations in Exhaled Breath Condensate differentiated LUAD and LUSC with high accuracy	AUC = 0.98, sensitivity = 100%, specificity = 88.0%	I 1 (4.8%) II 3 (14.3%) III 5 (23.8%) IV 12 (57.1%)	33572343 (https://www.mdpi.com/2075-4426/11/2/111)

**Table 5 ijms-24-13302-t005:** MiRNAs associated with LUAD prognosis.

MiRNA	Sample Source and Identification Effects	Statistics	Patient Stage, *n* (%)	References (PMID and Link accessed on 22 August 2023)
miR-125b	MiR-125b is decreased in LUAD tissues and correlates with poor prognosis	*p* = 0.001	/	35187068 (https://www.frontiersin.org/articles/10.3389/fmolb.2021.788690/full)
miR-142	Serum miR-142-3p is associated with early relapse in operable LUAD patients	*p* = 0.007	/	23410826 (https://www.lungcancerjournal.info/article/S0169-5002(13)00023-8/fulltext)
miR-145	MiR-145 level in LUAD tissues is an independent risk factor for both OS and DFS in LUAD	*p* = 0.004	I 34 (37.0%) II 25 (27.1%) III 33 (35.9%)	26582602 (https://www.nature.com/articles/srep16901)
miR-210	MiR-210 expression in LUAD tissues is a prognostic factor for OS in patients	*p* = 0.001	I 54 (67.5%) II-III 26 (32.5%)	25733977 (https://www.hindawi.com/journals/jo/2015/316745/)
miR-324	The combination of TP53 mutations and high miR-324-5p expression in LUAD tissues can predict poor prognosis	*p* = 0.052	/	34257080 (https://aacrjournals.org/mcr/article/19/10/1635/665704/MicroRNA-324-5p-CUEDC2-Axis-Mediates-Gain-of)
miR-650	MiR-650 expression level in LUAD tissues is significantly correlated with lymph node metastasis and clinical stage	*p* = 0.019	I-II 53 (55.2%) III-IV 43 (44.8%)	23991130 (https://journals.plos.org/plosone/article?id=10.1371/journal.pone.0072615)
miR-940	Decreased miR-940 expression in LUAD tissues can predict a negative prognosis in early-stage female patients	*p* = 0.011	IA 8 (65.7%), IB 4 (33.3%)	35004257 (https://tlcr.amegroups.org/article/view/58742/html)
miR-126	Both miRNAs within LUAD tissues exhibit the capability to predict pathological stage, tumor diameter, and lymph node metastasis.	AUC = 0.715, sensitivity = 64%, specificity = 75%	I 25 (51%) II-III 24 (49%)	27277197 (https://www.spandidos-publications.com/or/36/2/909)
miR-451a		AUC = 0.742, sensitivity = 84%, specificity = 67%		
miR-141	High miR-141 and miR-200c expression in LUAD tissues are associated with shorter OS through MET and angiogenesis	*p* = 0.009	I 94 (60.6%) II 34 (22%) III 27 (17.4%)	25003366 (https://journals.plos.org/plosone/article?id=10.1371/journal.pone.0101899)
miR-200c		*p* < 0.001		

**Table 6 ijms-24-13302-t006:** MiRNAs associated with LUAD therapy.

MiRNA	Identified Mechanism	References (PMID and Link accessed on 22 August 2023)
Tyrosine kinase inhibitors		
miR-1	decrease sensitivity to EGFR-TKI by changing tumor immune microenvironment	33305905 (https://onlinelibrary.wiley.com/doi/10.1002/cam4.3639)
miR-7	reduce EGFR expression in LUAD cell lines with acquired EGFR-TKI resistance	21712475 (https://aacrjournals.org/mct/article/10/9/1720/91084/Liposomal-Delivery-of-MicroRNA-7-Expressing)
miR-16	restoring sensitivity to erlotinib in KRAS-mutated LUAD in vitro and in vivo	34948154 (https://www.mdpi.com/1422-0067/22/24/13357)
miR-17	inhibit the EZH1 enhancer that contributes to EGFR-TKI resistance in cancer	27633093 (https://www.tandfonline.com/doi/full/10.1080/1061186X.2016.1207647)
miR-21	correlate with progression of EML4-ALK-translocated LUAD in patients prescribed ALK-TKI treatment	30658414 (https://www.mdpi.com/2072-6694/11/1/104)
miR-23a	inhibition of miR-23a increases the sensitivity of LUAD stem cells to erlotinib	28901474 (https://www.spandidos-publications.com/or/38/5/3064)
Chemotherapy—Cisplatin		
miR-10a	increase the cisplatin resistance of LUAD circulating tumor cells via targeting PIK3CA	32186774 (https://www.spandidos-publications.com/10.3892/or.2020.7547)
miR-15b	increase cisplatin resistance and metastasis by targeting PEBP4 in LUAD cells	25721211 (https://www.nature.com/articles/cgt201473)
miR-20a	suppress the PTEN/PI3K-AKT pathway to promote chemoresistance to cisplatin of LUAD cells	35857905 (https://onlinelibrary.wiley.com/doi/10.1002/ctm2.989)
miR-26a	responsible for A549 cell sensitivity in the treatment of cisplatin through E2F1-Akt pathway	26492332 (https://www.tandfonline.com/doi/full/10.1080/15384047.2015.1095405)
miR-30b	inhibit cancer progression and enhance cisplatin sensitivity in LUAD through targeting LRP8	33779882 (https://link.springer.com/article/10.1007/s10495-021-01665-1)
miR-31	inhibit cisplatin-induced apoptosis in LUAD cells by regulating the drug transporter ABCB9	24099915 (https://www.sciencedirect.com/science/article/abs/pii/S0304383513007039)
miR-32, miR-548a	promote sensitivity of LUAD cells to cisplatin by targeting ROBO1 and inhibiting the activation of Wnt/β-catenin axis	33854371 (https://www.ncbi.nlm.nih.gov/pmc/articles/PMC8039019/)
Chemotherapy—Celastrol		
miR-33a	enhance the sensitivity of LUAD cells to celastrol by regulating mTOR signaling	29484434 (https://www.spandidos-publications.com/ijo/52/4/1328)
Chemotherapy—Docetaxel		
miR-100	resensitize docetaxel-resistant LUAD cells (SPC-A1) by targeting Plk1	22120675 (https://www.sciencedirect.com/science/article/abs/pii/S0304383511007257)
Radiotherapy		
miR-15a/16	enhance radiation sensitivity of A549 cell by targeting the TLR1/NF-κB signaling	25442346 (https://www.redjournal.org/article/S0360-3016(14)04127-3/fulltext)
miR-18a	increase the radiosensitivity in LUAD cells via downregulating ATM and HIF-1α expressions	29860718 (https://onlinelibrary.wiley.com/doi/10.1002/cam4.1527)
miR-26b	downregulate ATF2 to enhance radiosensitivity of LUAD cells	32476275 (https://onlinelibrary.wiley.com/doi/10.1111/jcmm.15402)
miR-29b	upregulation of VANGL1 by IGF2BPs and miR-29b attenuates the detrimental effect of irradiation on LUAD	33228740 (https://jeccr.biomedcentral.com/articles/10.1186/s13046-020-01772-y)

## Data Availability

Data sharing not applicable. No new data were created or analyzed in this article.

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
