# Peer review of "MiRNAs in Lung Adenocarcinoma: Role, Diagnosis, Prognosis, and Therapy"

_ijms, 2023, doi:10.3390/ijms241713302_

Round 1

Reviewer 1 Report

It is an undeniable fact that lung cancer is a major public health problem and remains the leading cause of cancer death worldwide. Among the various types of lung cancers, adenocarcinoma of the lung is the most common form. Today, the study of miRNAs (miRNAs) is a popular research topic, they have been shown to play a critical role in gene regulation, and their involvement in cancer development has been extensively investigated. This review focuses specifically on miRNAs, especially in LUAD, which fills one of the gaps in this area. The review is detailed, well planned, but there are small flaws, in particular

1. In the tables, it is better to provide a link to the list of references to the article in addition to the number in PubMed.

2. In the section on the diagnostic capabilities of miRNAs, it is necessary to give a summary table of sensitivity and specificity (or AUC-ROCK) for individual miRNAs, and also indicate at what sample and at what stages these values were established.

3. Similarly for the prognostic role of miRNAs (see item 2).

Author Response

Response to Reviewer 1 Comments

Dear reviewer

Thank you sincerely for your comments on our article. Your feedback, which encompasses both commendations and areas for improvement, is greatly appreciated. Your input has enhanced the clarity and comprehensiveness of our work. We have made revisions to the manuscript. Below, we provide explanations that account for the changes we implemented based on your comments.

Thank you very much!

Best wishes

Song

Point 1: In the tables, it is better to provide a link to the list of references to the article in addition to the number in PubMed.

Response 1: Thank you for your suggestion. We have added links to references in the tables, highlighted in blue font.

Point 2: In the section on the diagnostic capabilities of miRNAs, it is necessary to give a summary table of sensitivity and specificity (or AUC-ROCK) for individual miRNAs, and also indicate at what sample and at what stages these values were established.

Response 2: Thank you for your valuable suggestion. We have incorporated sensitivity and specificity (or AUC-ROCK) statistics into Table 4, along with clear indications of the corresponding samples and stages. These additions have been highlighted in blue font.

Point 3: Similarly for the prognostic role of miRNAs (see item 2).

Response 3: Thank you for your suggestion. We have added statistics and indicated the corresponding samples and stages to Table 5. These additions also have been highlighted in blue font.

Reviewer 2 Report

The authors reviewed in detail the data available in the literature on the role of miRNAs associated with LUAD.
This material is very useful to clinicians and will make it easier for researchers to understand the advances made in miRNA
research at LUAD.  

Author Response

Response to Reviewer 2 Comments

Dear reviewer

Thank you very much for your recognition of our article!

Best wishes

Song

Reviewer 3 Report

The manuscript „MiRNAs in lung adenocarcinoma: role, diagnosis, prognosis and therapy “, written by Song Y, Kelava L and Kiss I. presents the roles of miRNAs in lung adenocarcinoma. The manuscript describes types of lung carcinoma, production of miRNA and different miRNAs in lung carcinoma, divided on tumor suppressors and oncogenes, miRNAs in diagnostics of lung adenocarcinoma, their roles in prognostics and possible therapy.

The manuscript is well written, comprehensive, detailed and contains a huge amount of data. Additional data are presented in supplementary tables. References are up to date.

There are only some minor comments:

Jagged 2 is mentioned in two paragraphs, once as JAG2, and then as Jagged, these data could be put together.

Several abbreviations do not have explanation (lines 401, 417, 500)

Paragraph describing extracellular vesicles could be placed in the part describing roles or diagnostics of miRNAs in adenocarcinoma.

References in tables could be cited as numbers of references, not PMID numbers (they are specific for PubMed).

Sentence reorganization or explanation: lines 181-182, 237

Explanation of „poor mortality“

English language is fine.

Author Response

Response to Reviewer 3 Comments

Dear reviewer

Thank you sincerely for your comments on our article. Your feedback, which encompasses both commendations and areas for improvement, is greatly appreciated. Your input has enhanced the clarity and comprehensiveness of our work. We have made revisions to the manuscript. Below, we provide explanations that account for the changes we implemented based on your comments.

Thank you very much!

Best wishes

Song

Point 1: Jagged 2 is mentioned in two paragraphs, once as JAG2, and then as Jagged, these data could be put together.

Response 1: Thank you for your suggestion. We have put these data together and highlighted in green font.

Point 2: Several abbreviations do not have explanation (lines 401, 417, 500)

Response 2: Thank you for your suggestion.

The abbreviations "SVM" and "TSP" are introduced in line 401. In accordance with the editor's suggestion, we have rewritten this sentence. The new sentence omits these two abbreviations.

The abbreviation used in line 417 is "EV‐derived." We have included an explanatory note: "extracellular vesicle (EV)‐derived."

The abbreviation employed in line 417 is "TCGA." We have provided an note: "The Cancer Genome Atlas."

These additions have been highlighted in green font.

Point 3: Paragraph describing extracellular vesicles could be placed in the part describing roles or diagnostics of miRNAs in adenocarcinoma.

Response 3: Thank you for your suggestion. Exosome-derived miRNAs primarily serve for diagnosing LUAD thus we have incorporated them into the diagnostic section. We have also highlighted the title of this section, 'Exosomes,' in green font.

Point 4: References in tables could be cited as numbers of references, not PMID numbers (they are specific for PubMed).

Response 4: Thank you for your suggestion. Another reviewer also raised this point. Their suggestion was to consider including links to the list of references in addition to the PubMed number. We've introduced links as we felt they would offer readers a more convenient way to access the content directly.

Point 5: Sentence reorganization or explanation: lines 181-182, 237

Response 5: Thank you so much for your thorough review! Due to our oversight, there are errors in both sentences. Your corrections are greatly appreciated. We have since rephrased these two sentences and highlighted them in green font.

Point 6: Explanation of “poor mortality”

Response 6: Thank you once again for your comprehensive review! The use of "poor mortality" is not accurate in this context. "Worse mortality" is the appropriate choice. We have rectified this error and highlighted the correction in green font.

Round 2

Reviewer 1 Report

The authors took into account the weight of the comments of the reviewers and significantly revised the manuscript. I believe that the present form of the manuscript can be accepted for publication